# Combination of Multiple Microsatellite Analysis and Genome-Wide SNP Genotyping Helps to Solve Wildlife Crime: A Case Study of Poaching of a Caucasian tur (*Capra caucasica*) in Russian Mountain National Park

**DOI:** 10.3390/ani11123416

**Published:** 2021-11-30

**Authors:** Andrey Rodionov, Tatiana Deniskova, Arsen Dotsev, Valeria Volkova, Sergey Petrov, Veronika Kharzinova, Olga Koshkina, Alexandra Abdelmanova, Anastasia Solovieva, Alexey Shakhin, Nikolay Bardukov, Natalia Zinovieva

**Affiliations:** L.K. Ernst Federal Research Center for Animal Husbandry, Dubrovitsy, Podolsk Municipal District, Moscow Region, 142132 Podolsk, Russia; rodiand@yandex.ru (A.R.); moonlit_elf@mail.ru (V.V.); citelekle@gmail.com (S.P.); veronika0784@mail.ru (V.K.); olechka1808@list.ru (O.K.); preevetic@mail.ru (A.A.); anastastasiya93@mail.ru (A.S.); alexshahin@mail.ru (A.S.); bardukv-nikolajj@mail.ru (N.B.); n_zinovieva@mail.ru (N.Z.)

**Keywords:** poaching, wild goat, single-nucleotide polymorphisms, microsatellites, evidence, DNA typing

## Abstract

**Simple Summary:**

DNA molecular techniques, including multiple microsatellite analysis and genome-wide SNP-genotyping, were used to unlock and prove the poaching of wild goats (*Capra caucasica*) in an area of the Caucasian mountains in Russia.

**Abstract:**

Poaching is one of the major types of wildlife crime in Russia. Remnants of goats (presumably the wild endemic species, the Caucasian tur) were found in an area of the Caucasian mountains. The case study involves a suspected poacher whose vehicle was found to have two duffel bags containing pieces of a carcass, which he claimed was that of a goat from his flock. The aim of the forensic genetic analysis for this case was to (i) establish individual identity and (ii) perform species identification. DNA typing based on fourteen microsatellites revealed that STR-genotypes generated from pieces of evidence found at crime scene fully matched those obtained from the evidence seized from the suspect. The results of genome-wide SNP-genotyping, using Illumina Goat SNP50 BeadChip, provided evidence that the poached animal was a wild Caucasian tur (*Capra caucasica*). Thus, based on comprehensive molecular genetic analysis, evidence of poaching was obtained and sent to local authorities. To our knowledge, this case study is the first to attempt to use DNA chips in wildlife forensics of ungulates.

## 1. Introduction

The wide application of DNA methods in human forensic investigation has led to significant successes in animal evidence analysis by the horizontal transfer of molecular and statistical techniques [1]. Animal forensic genetics serves to unlock and provide evidence for crimes against domestic species and wildlife [2] and is generally based on the common principles of human forensics. Thus, DNA typing using microsatellites or short tandem repeats (STR) is the most powerful and robust technique for matching individual specimens to other pieces of evidence [1]. 

However, animal forensic genetics has several specific features including frequent necessity for species identification. Often, testing for species identification is a crucial procedure in establishing wildlife poaching cases [3,4,5]. Due to several wildlife species having “relatives” (ungulates, boars, etc.) among domestic animals, a suspected poacher could claim that retrieved biological objects belonged to livestock species, which were his own property [4]. For this reason, various types of DNA markers, including nuclear [3,4,5] and mitochondrial markers [6], are used for species assignment. A combination of different types of DNA markers provides higher resolution and accuracy; this facilitates correct species and origin determination for unlocking wildlife cases, in comparison with using only one type of DNA marker. A combination of the mitochondrial Cytochrome-b gene (*Cyt-b*) with 12 microsatellite loci was beneficial and allowed a successful investigation of poaching of the Cypriot mouflon (*Ovis orientalis ophion*), wherein three shot mouflon were discovered, and bloodstained items were seized from a suspect’s vehicle [5]. Use of a porcine-specific STR system and real time PCR-based assays of single nucleotide polymorphisms in the *NR6A1* and *MC1R* genes was proposed for discrimination of wild boars from domestic pigs, and validated for forensic purposes [7].

Wildlife poaching happens worldwide and leads to a significant reduction in animal biodiversity that is especially dramatic for threatened and endangered species. Poaching topped the list of crimes committed in Russia’s forestry sector in 2021 [8], and cases have risen further during the COVID-19 pandemic [9]. Unfortunately, many poachers do not face charges due to lack of evidence. For this reason, the strength of DNA evidence is becoming more relevant, as a means to protect threatened species and prevent future crimes. 

Caucasus Mountain Chain is the habitat of the Caucasian tur (*Capra caucasica*). This local endemic ungulate is subdivided into three ecotypes: West-Caucasian, East-Caucasian, and Mid-Caucasian. During the hunting season, which lasts from 1 August to 30 November [10], hunting turs is legal with a valid, official hunting license. 

Here, we report a case involving the poaching of a wild Caucasian tur in the Republic of Kabardino-Balkaria in Russia in 2020. In February 2020 (which is out of hunting season), goat remnants including visceral organs and bones with meat were found in the gorge in the Elbrus district of the Republic of Kabardino-Balkaria, a region in the Caucasian Mountains of Southern Russia. According to Federal Law of the Russian Federation “On hunting and preserving hunting resources”, local authorities considered this offense a violation of the hunting rules (poaching). Furthermore, two duffel bags containing pieces of carcass (meat and bones) were seized by local police officers from a suspected poacher, who claimed that the retrieved items belonged to domestic and hybrid goats from his flock. However, his claim could not be proved or refuted without further molecular forensic analyses. 

Thus, we provide the results of the molecular genetic analysis for this case. Using a combination of DNA approaches, including multiple STR- and genome-wide SNP-genotyping, we aimed to: (1) establish the links between the items found at the crime scene and pieces of carcass retrieved from a suspect`s vehicle, and (2) identify the species of the illegally harvested animal. 

Moreover, we demonstrate the usefulness of the combination of microsatellites and SNP-genotyping (via DNA chips) to track and prove wildlife crimes committed in the Russian National Park in the Caucasian Mountains. 

## 2. Materials and Methods

### 2.1. Pieces of Evidence 

Pieces of evidence for this criminal offense included seven biological items which were found at the crime scene and were seized from a suspect’s vehicle (Figure 1). 

A detailed description of the investigated biological samples is presented in Table 1.

### 2.2. DNA Extraction 

DNA isolation was performed from the studied samples using commercial DNA-Extran-2 kits (CJSC Syntol, Russia) according to the manufacturer’s recommendations. The concentrations of double-stranded DNA were measured on a Qubit 4.0 fluorimeter (Thermo Fisher Scientific (formerly Life Technologies), Wilmington, DE, USA). The DNA purity was checked by evaluating the absorption ratio of OD260/OD280 on a NanoDrop2000 spectrophotometer (Thermo Fisher Scientific, Wilmington, DE, USA).

### 2.3. Microsatellite Analysis and Sex Determination

In total, fourteen goat microsatellite loci including INRA005, ILSTS19, SRCRSP8, CSRD247, ILSTS87, SRCRSP5, SRCRSP23, MAF65, ILSTS008, MCM527, INRA006, INRA063, OARFCB20, and ILSTS011, which were recommended by the International Society of Animal Genetics (ISAG) [11] and were combined into three multiplexes, were used in this study. The forward oligonucleotide primer from each pair was labeled with one of the fluorescent dyes (R6G, HEX, or FAM). 

PCR was carried out in a 9 µL reaction volume, containing 1 µL of genomic DNA template, 1 µL 10 × PCR-buffer, 1 µL dNTPs (1 mmol/L), 0.5 µL MgCl_2_, 3.6 µL of primer master mix (20 pmol), and 0.1 µL of Smart Taq DNA polymerase (Dialat ltd, Moscow, Russia). 

The conditions for PCR were as follows: initial denaturation of 10 min at 95 °C, followed by 40 cycles of 30 s at 95 °C, 40 s at annealing temperature at 58 °C, and 1 min at 72 °C; and a final extension of 10 min at 72 °C. 

Fragment analysis was performed on an ABI3130xl genetic analyser (Applied Biosystems, Beverly, MA, USA) using GeneScan™-350 ET ROX as a fragment standard. The Gene Mapper software v. 4 (Applied Biosystems, Beverly, MA, USA) was used to determine the allele lengths.

The statistical power of the chosen set of microsatellite loci was tested by estimation of the Probability of Identity (PI) or Match Probability and Probability of Identity for Sibs (PIsibs) in 408 samples from nine goat breeds, including Orenburg (*n* = 30), Soviet Mohair (*n* = 30), Dagestan Local (*n* = 28), Dagestan Downy (*n* = 27), Saanen (*n* = 34), Murciano-Granadina (*n* = 37), Karachaev (*n* = 159), Altai White Downy (*n* = 32), and Altai Mountain (*n* = 31).

The program GenAlEx 6.503 [12,13] was used to estimate PI and PIsib values following the equation [14],
PI = 2(**Σ***p*^2^*_i_*)^2^ − **Σ***p*^2^*_i_*, for each locus 
where *p_i_* is the frequency of the *i*th allele at a locus.

For sex determination, the fragment of the amelogenin gene (AMEL) was amplified using a pair of primers, SE47 (forward: 5′-CAGCCAAACCTCCCTCTGC-3′) and SE48 (reverse: 5′-CCCGCTTGGCTTGTCTGTTGC-3′), flanking region with a specific deletion on the Y chromosome. The analysis was performed according to the method proposed by Petrov S. et al. [15]. Sex determination is based on the number of detected peaks; females have single peak with allele size of 264 bp (XX, homozygote genotype), and males have two peaks with allele sizes of 264 and 218 bp (XY, heterozygous genotype).

### 2.4. Single Nucleotide Polymorphisms Genotyping and References Groups 

Sample No. 2 (the X sample) was genotyped using the Illumina Goat SNP50 BeadChip containing 53,347 SNPs [16,17]. To perform species assignment, we constructed the combined data set, which included SNP genotypes of the X sample, domestic goats, and wild Caucasian tur. As domestic goat representatives, Saanen and Karachaev breeds were selected as they are the most popular goat breeds in the Republic of Kabardino-Balkaria. Samples of goats from Saanen (*n* = 5) and Karachaev breeds (*n* = 5) were genotyped earlier [18]. Samples of Caucasian turs comprised of East-Caucasian tur ecotype from Dagestan (E_TUR, *n* = 5), West-Caucasian tur ecotype from Karachay-Cherkessia (W_TUR, *n* = 5), and Mid-Caucasian tur ecotype from Kabardino-Balkaria (M_TUR, *n* = 5), which were genotyped for previous study [19]. 

### 2.5. Data Analysis 

SNP quality filtering was performed in PLINK 1.9 [20]. 

After quality control, 43,726 variants passed the filters (only SNPs with known position, located on autosomes with a minor allele frequency of 1%, and successful genotyping in at least 90% of individuals). This set of SNPs was used for multilocus heterozygosity (MLH) analysis. 

For PCA, Admixture and Neighbor-Net, LD pruning was performed (indep-pairwise 50 5 0.5). After LD pruning, 7980 SNPs were selected.

Principle component analysis (PCA) was performed with PLINK 1.9 (pca 4) and visualized in the R package “ggplot2” [21]. An individual tree based on the pairwise identity-by-state (IBS) distance matrix (distance 1-ibs) was constructed using the Neighbor-Net algorithm implemented in SplitsTree 4.14.6 [22]. 

To estimate and visualize the distribution of heterozygosity at the individual level, multilocus heterozygosity (MLH) was calculated in the R package “inbreedR” [23]. Cluster analysis was performed in Admixture 1.3 software [24] and visualized in the R package “pophelper” [25].

The positions on chromosomes in this paper correspond to the reference genome of a domestic goat, because the Illumina Goat SNP50 BeadChip was developed for domestic goats (*Capra hircus*).

## 3. Results

### 3.1. Identity Analysis Based on Microsatellites Markers 

Using a commercial kit, we were able to extract genomic DNA of proper quality from six of seven studied samples (No. 1–6). The concentrations of double-stranded DNA varied from 5.6 to 444.0 ng/µL and OD260/OD280 values were 1.46–1.92 (Appendix A). No detectable amount of DNA (0.1 ng/µL) was obtained from sample No. 7 (washout of a brown stain on the gauze pad); therefore, this sample was excluded from further analysis. 

The used set of fourteen microsatellites was tested by calculating PI values in nine goat breeds which are reared in Russia. PI values for unrelated individuals in different populations for fourteen microsatellites varied from 6.1 × 10^−13^ in the Murciano-Granadina breed to 8.3 × 10^−17^ in the Karachaev breed (Appendix A). These estimates correspond to a high statistical power of the applied marker set.

Multiple microsatellite genotypes were generated for all six samples. Four samples (No. 1, No. 2, No. 3, and No. 5) were successfully genotyped for all of fourteen loci, while genotypes for the two remaining samples (No. 4 and No. 6) were produced for thirteen loci, excluding SRCRSP8. The genotypes, which were obtained for the items seized from a poacher (samples No. 1–4), fully matched the microsatellite genotypes, which were generated for the items found on the crime scene (No. 5–6) (Table 2). 

The sex determination analysis showed that all the studied samples had an identical heterozygous genotype 218/264 of the AMEL gene, which corresponds to the male sex (Table 1).

Thus, based on microsatellite analysis and amelogenin genotypes, we may conclude that all analyzed items belong to a single animal. 

### 3.2. Species Assignment Based on SNP Genotyping

Principal component analysis (PCA) performed for the X sample, domestic goats (*Capra hircus*), and Caucasian turs (*Capra caucasica*) (Figure 2A) showed that the first principal component (PC1), which was accounted for 8.7% of genetic variability, clearly separated wild and domestic goats. The X sample was closely clustered with Caucasian tur individuals. Therefore, to establish that the X sample belonged to one of the Caucasian tur ecotypes, we performed the alternative PCA without domestic goats (Figure 2B). Therefore, the X sample joined the group of Mid-Caucasian tur ecotype, which was separated from the East-Caucasian tur ecotype by PC1 (2.55% of genetic variability) and from the West-Caucasian tur ecotype by PC2 (1.98% of genetic variability). 

The results provided by Admixture analysis (Figure 2C) corresponded with those obtained by PCA. At K = 2, the X sample demonstrated a common genetic background with Caucasian turs (green color). At K = 5, the investigated sample showed a shared ancestry with individuals from the Mid-Caucasian tur ecotype (aquamarine color). 

A Neighbor-Net dendrogram (Figure 2D) demonstrated the presence of two large clusters: the first one included the X sample and wild Caucasian turs, and the second one consisted of domestic goats. Within the “wild” cluster, the X sample joined the branch, which was formed by individuals of the Mid-Caucasian tur ecotype. 

In addition, we calculated values of individual multilocus heterozygosity (MLH) in the X sample, as well as in the groups of wild and domestic goats (Figure 3). The mean values of multilocus heterozygosity were lower in Caucasian tur ecotypes (from 0.0297 ± 0.0007 in E_TUR to 0.0353 ± 0.0002 in W_TUR) than the values identified in domestic goat breeds (0.3919 ± 0.0034 in KARA and 0.4196 ± 0.0028 in SAAN). The value of individual multilocus heterozygosity in the X sample was 0.0353 and was similar to that found in Caucasian tur ecotypes. This provides additional evidence of the studied sample belonging to the wild Caucasian tur species.

Thus, our results, obtained by several complementary approaches, clearly demonstrated that the X sample belongs to the Caucasian tur, particularly to the Mid-Caucasian tur ecotype from the Kabardino-Balkarian habitat. 

## 4. Discussion

Wildlife crimes cause huge damage in animal species diversity, consequently reducing their genetic variability and ability to face unpredictable environmental and anthropogenic changes. The most tragic outcome results in the extinction of threatened species. Along with effective protection measures and conservation programs, proving the offences of perpetrators and enforcing punishment are crucial to control and prevent future wildlife crimes [26]. 

Poaching—the hunting of protected species outside of the hunting season without scientific license or valid permissions issued by the Ministry of Natural Resources and Environment—is strictly prohibited and is classified as a criminal offense in the Russian Federation. However, procedures for wildlife forensics as well as domestic animal forensic genetics are still lagging behind human forensics. Due to a lack of clear instructions or recommendations regarding the use of nonhuman DNA in forensic genetic investigations, perpetrators often avoid prosecution for poaching and other wildlife crimes. 

In view of this, creating precedents for tracking and proving wildlife crimes using DNA markers, with a focus on their resolution power, is an important step in protecting endangered species and preventing future crimes. 

Molecular genetic investigation for this case included two steps. The first step was to prove (or refute) the identity of the pieces of evidence found at the crime scene and those seized from the suspect. For this purpose, we used three multiplex panels of microsatellites. 

Microsatellite analysis provides key evidence for wildlife crime investigations in cases of poaching, the identification of stolen animals, and the authentication of legally traded wildlife products [1,2]. Specifically, microsatellites showed their effectiveness in cases of poaching that occurred in Italy, under similar circumstances to those presented in this paper. In the first case, a wild boar sow was slaughtered with a knife in a national park in Northern Italy [3]. The suspect was convicted of wild boar poaching based on matching microsatellite profiles obtained from blood on the knife and a wild boar carcass [3]. The second case of poaching, on the island of Sardinia, was proved based on the establishment of a connection between the carcass of a mouflon and items seized from a vehicle [4]. 

Due to the high informativity and robustness of microsatellites as well as their lower cost, we preferred to use microsatellites to provide evidence of the single origin of the samples found at the crime scene and items seized from the suspect.

The second step of our investigation was species identification of the studied sample. Species assignment is the most frequent and difficult aspect of wildlife forensics [2], for which the effectiveness and robustness depend on the choice of the used DNA marker. 

Mitochondrial genes are reported to be highly informative in species testing [6]. Furthermore, the cytochrome c oxidase I (*COI*) mitochondrial gene is considered as the Barcode for Life Consortium [2,27]. DNA barcoding technology has been used in various cases involving ungulate species. For example, twenty species of antelope, buffalo, and domestic Bovidae were correctly clustered based on short fragment *COI* sequences [28]. Chen, J. et al. [29] used mini-barcoding technology based on sequencing of mitochondrial gene *COI* for successful differentiation of Saiga antelope (*Saiga tatarica*) from another ungulate species. 

However, the use of mitochondrial genes may not provide a detectable resolution between closely related taxa [30] and may be misleading in the presence of hybrids, whereby the tested sample will be wrongfully assigned to the female parental species [31]. 

In our study, we did not consider using mitochondrial DNA sequencing due to specific features of the geographical location of the crime scene. The Republic of Kabardino-Balkaria, where the present crime was committed, is in the mountain terrain of the Caucasian Mountains. This area is a hybrid zone which is shared by domestic goats raised by local people and wild goats (turs) that historically dwell on the mountain slopes. Introgression between the species most frequently results from the interaction between wild males and domestic females. However, the opposite situation is also possible. Young, wild females are sometimes taken by local hunters and herdsmen and put into domestic herds.

Single-nucleotide polymorphism (SNP) markers were also successfully applied to species differentiation. For example, a panel of 12-ancestral informative SNP implemented in the SNaPshot multiplex genotyping system was used to discriminate Mongolian wolves from domestic dogs [30]. However, the proposed approach should be validated on various wolf and dog populations to be recommended for routine testing in forensic genetics. 

Increasing the number of genotyped SNP markers raised the accuracy of genetic analysis. In human studies, high-density SNP-genotyping microarrays showed significant resolution power for detecting individual DNA in a complex genomic DNA mixture [32]. In addition, the implementation of high-throughput DNA-genotyping arrays has led to the generation of SNP genotypes of different domestic and wild animal species, which might be used as reliable comparison groups for taxonomic, species, and breed/ecotype assignment of the tested samples, without further validation. Thus, Yang Z et al. [33] demonstrated the utility of the Illumina Canine HD BeadChip for dog breed recognition, which is crucial in domestic animal forensic genetics. 

Furthermore, DNA chips are found to be highly informative for wild species whose domestic relatives’ DNA chips are available. Thus, the application of a medium-density OvineSNP50 BeadChip allowed the differentiation of the European mouflon (*Ovis aries musimon*) from 14 domestic sheep breeds, as well as the detection of introgression events between these species in some populations [34]. A medium-density DNA array was able to discriminate different geographical groups of wild show sheep (*Ovis nivicola*) in Siberia [35]. In addition, medium-density DNA chips were used to recognize hybrids originating from crossings between domestic sheep and wild argali (*Ovis ammon*) [36].

In our study, we chose the commercial DNA chip (Illumina Goat SNP50 BeadChip), which was created for domestic goats. This array was successfully used in taxonomy studies of Caucasian turs [19], which are used as comparison groups in the present work. As expected, we not only successfully established species assignment but also found the group/ecotype origin. Furthermore, the studied X sample as well as other wild turs from comparison groups displayed lower levels of genetic diversity than domestic goats; this is consistent with the findings detected in mouflon and domestic sheep [34]. 

Novel DNA technologies should be verified for forensic applications, as a wide range of methods can be confusing for forensic practitioners [37]. To our knowledge, the present work is the first attempt to use DNA chips in wildlife forensics of ungulates. The results demonstrate the significance of implementing novel molecular genetic solutions to untangle cases in which the main pieces of evidence are presented as biological material. 

## 5. Conclusions

In this study, the combination of two types of markers, namely microsatellites and SNP-genotyping array (DNA chip), allowed us to determine the corpus delicti in an accurate manner and provided molecular genetic evidence of poaching of the Caucasian tur. 

Based on multiplex microsatellite analysis, matches between items found at the crime scene and those which were seized from the suspect were established. Along with reliable species identification, the application of a genome-wide SNP array allowed the determination of the definite ecotype (Mid-Caucasian tur), whose habitat covers the Republic of Kabardino-Balkaria. 

We believe that our results will be useful for tracking and proving wildlife crimes and might be considered as a precedent in wildlife forensics in Russia. 

## Figures and Tables

**Figure 1 animals-11-03416-f001:**
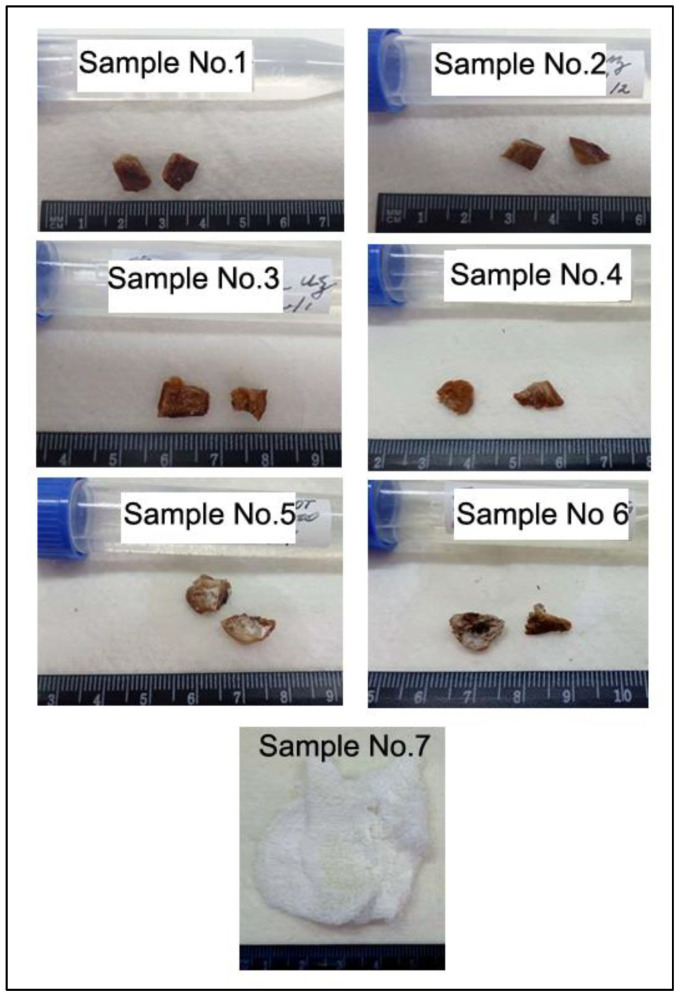
Seven items seized from a vehicle (samples 1–4) and found on crime scene (samples 5–7) which are assumed as pieces of evidence for this wildlife crime.

**Figure 2 animals-11-03416-f002:**
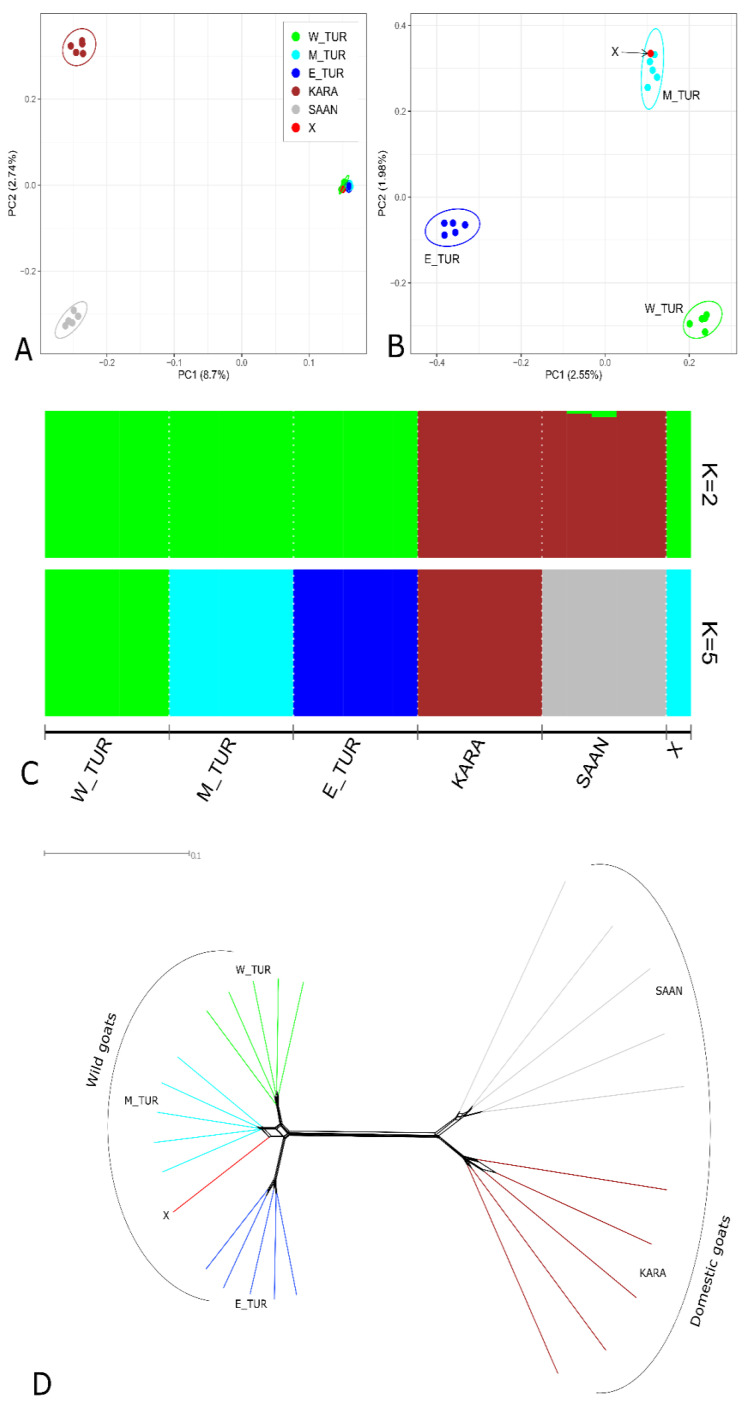
Identification of the species assignment of the X sample based on SNP genotyping: (**A**) principal component analysis (PCA) of the X sample, domestic goats (*Capra hircus*) and Caucasian tur (*Capra caucasica*); (**B**) principal component analysis (PCA) of the X sample and Caucasian turs without domestic goats; (**C**) admixture analysis of the X sample, domestic goats, and Caucasian tur at K = 2 and K = 5; (**D**) neighbor-Net dendrogram of the X sample, domestic goats and Caucasian tur, based on IBS distances. W_TUR = West-Caucasian tur ecotype, E_TUR = East-Caucasian tur ecotype, M_TUR = Mid-Caucasian tur ecotype, KARA = Karachaev goat breed, SAAN = Saanen goat breed.

**Figure 3 animals-11-03416-f003:**
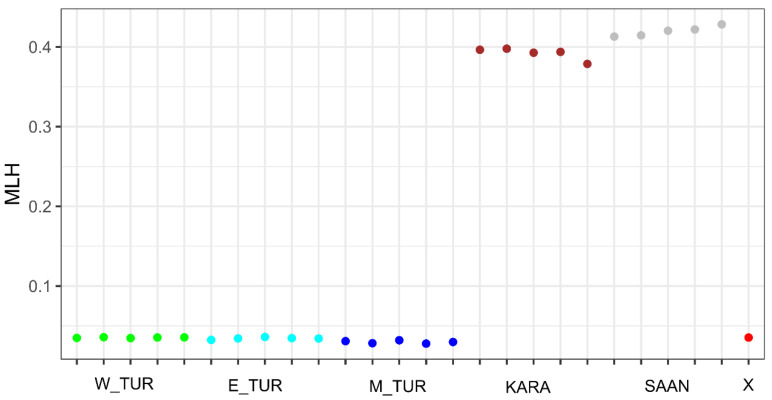
Individual multilocus heterozygosity (MLH) in the Caucasian tur (*Capra caucasica*) ecotypes, domestic goat (*Capra hircus*) breeds, and in the X sample. W_TUR = West-Caucasian tur ecotype, E_TUR = East-Caucasian tur ecotype, M_TUR = Mid-Caucasian tur ecotype, KARA = Karachaev goat breed, SAAN = Saanen goat breed.

**Table 1 animals-11-03416-t001:** Description of the biological items, which are pieces of evidence for this forensic investigation.

Items	Location	Description
Sample No. 1	Suspect’s vehicle	Muscle tissue: a fragment from the first piece of meat from duffel bag No. 1
Sample No. 2	Suspect’s vehicle	Muscle tissue: a fragment from the second piece of meat from duffel bag No. 1
Sample No. 3	Suspect’s vehicle	Muscle tissue: a fragment from the first piece of meat from duffel bag No. 2
Sample No. 4	Suspect’s vehicle	Muscle tissue: a fragment from the second piece of meat from duffel bag No. 2
Sample No. 5	Crime scene	Muscle tissue: a fragment from the piece of meat
Sample No. 6	Crime scene	A viscera- fragment
Sample No. 7	Crime scene	Washout of brown stain on the gauze pad:

**Table 2 animals-11-03416-t002:** Microsatellite genotypes of studied items for fourteen loci.

Loci			Samples			
	No. 1	No. 2	No. 3	No. 4	No. 5	No. 6
INRA005	114/118	114/118	114/118	114/118	114/118	114/118
ILSTS19	169/169	169/169	169/169	169/169	169/169	169/169
SRCRSP8	212/212	212/212	212/212	-	212/212	-
CSRD247	239/243	239/243	239/243	239/243	239/243	239/243
ILSTS87	135/137	135/137	135/137	135/137	135/137	135/137
SRCRSP5	170/170	170/170	170/170	170/170	170/170	170/170
SRCRSP23	82/82	82/82	82/82	82/82	82/82	82/82
MAF65	113/113	113/113	113/113	113/113	113/113	113/113
ILSTS008	178/180	178/180	178/180	178/180	178/180	178/180
MCM527	171/171	171/171	171/171	171/171	171/171	171/171
INRA006	103/111	103/111	103/111	103/111	103/111	103/111
INRA063	177/177	177/177	177/177	177/177	177/177	177/177
OARFCB20	97/99	97/99	97/99	97/99	97/99	97/99
ILSTS011	283/283	283/283	283/283	283/283	283/283	283/283
AMEL	XY (218/264)	XY (218/264)	XY (218/264)	XY (218/264)	XY (218/264)	XY (218/264)

## Data Availability

SNP genotypes of the Caucasian tur are available on reasonable request after signing a research (MTA) agreement.

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
