# Peer review of "Combination of Multiple Microsatellite Analysis and Genome-Wide SNP Genotyping Helps to Solve Wildlife Crime: A Case Study of Poaching of a Caucasian tur (Capra caucasica) in Russian Mountain National Park"

_animals, 2021, doi:10.3390/ani11123416_

Round 1

Reviewer 1 Report

Dear authors,

In my opinion the article has some aspects that must be answered prior the consideration to be published in Animals.

1.- In the research a total of 14 microsatellites have been used to check that two different samples belongs to the same animal and after that the species classification of the sample has been made using SNPs. Why the authors has changed the molecular markers used?

Microsatellites has been traditionally used in paternity, traceability and forensic analysis. But, the authors did not explain how they choose the microsatellites (this is crucial in this kind of analysis, ISAG recommendations?) and also did not estimate some important parameters as the probability that two random samples have the same genotype. If it is not possible to estimate such parameter and the microsatellites are the same or very similar to ISAG recommendations perhaps they can add some references in which the parameter has been estimated and evidence that the microsatellites genotyped are suitable for the purpose of the article.

Furthermore, SNPs has been used in goats for forensic, paternity and traceability analysis, so as I mentioned before, please explain why the molecular markers has been changed. Could the authors achieved that the samples with different locations belongs to the same animal using SNPs? If yes, why they used microsatellites (the samples were genotyped with SNPs as evidenced reference 15)?

2.- The number of samples of each group is low (5) but a reference of the article with most of the authors (reference 15) showed that there are available a higher number of samples to be genotyped with SNPs for domesticated goat breeds (Karachaev and Saanen). The authors can explain how the low number of samples used could affected the results. Also, the multilocus heterozygosity (MLH) estimated in reference 15 (>0.22) for the Caucasian Tur are different from that showed in the article (<0.1) while the samples are the same, how can the authors explained this? Or perhaps I am wrong.

3.- As the authors mentioned in the introduction, the owner of the samples said that the samples are from goats of their flock or hybrids. What is/are the breed of the flock? The authors did not explain the reasons to include samples of two goat breeds Karachaev and Saanen, why not others breeds? There is no explanation in the article about how the results could change with hybrids samples.

4.- Finally, in the conclusion section it is mentioned the successfull combination of microsatellite and SNPs markers. But it is not clear the benefits to use it together instead each one alone. There are many cases in which microsatellite or SNPs have been used alone, but if we used together what are the benefits?

Best regards.

Author Response

Dear Reviewer,

thank you very much for the valuable comments!

1. In this study case, there were different samples which were found at the crime scene and were seized from a suspect's vehicle. Firstly, we required the genetic evidence that all samples belonged to one animal and further to identify the species assignment. We achieved the first goal with using microsatellites which are traditionally used in paternity, traceability and forensic analysis as well as characterized by lower cost compared with using SNP chips for all six samples. In case we had got different individual microsatellite profiles for studied samples, no further investigation was needed because of lack of criminal offense.

Thus, we used microsatellites to verify an identity of the samples found at crime scene with those, which were seized from a suspect poacher.

We chose microsatellites, which were recommended by the International Society of Animal Genetics (ISAG) for Caprine. According to reviewer suggestions, we tested the selected set of microsatellites by estimating the probability of identity (PI). PI values for unrelated individuals in different populations for fourteen microsatellites varied from 6.1 × 10-13 to 8.3 ×10-17. These estimates correspond to a high statistical power of the applied marker set. We added relevant information in the Materials and Methods (P4L126-130), and Results (P5L287-290) sections in the main text.

As we established the belonging of all samples to one individual, we used SNP, because we only need to genotype one sample. It was inexpensive, but the accuracy of the analysis was much higher that is essential for species assignment. Besides we had reference SNP-genotypes for 15 turs, represented by three ecotypes, which were generated for earlier published article [reference #19 in the revised version]. To avoid bias, we added comparable number of samples of domestic goats, represented by two breeds which are the most popular in the area (five samples for each of two breeds n= 10).

2. Sample for this study included 15 turs, and ten samples of domestic goats which are quite sufficient to established whether studied sample belonged to turs and not to domestic goats. The difference between wild and domestic goat species is quite large. The number of polymorphic loci on a chip for any of the domestic goat breeds will be several tens of thousands of loci, because they all originated from one species (the bezoar goat). The tur has lower number of polymorphic loci (~ 5 thousand).

Increasing the sample size will not affect the results in this aspect.

The difference in multilocus heterozygosity is due to fewer SNPs were used in the paper [reference #19 in the revised version]. In that paper, only loci, which were polymorphic for the tur (~ 5 thousand) were used for the analysis to compare groups of turs with each other. In present article, all SNPs from the chip were used, since the aim was to determine if the sample under study is Capra hircus or Capra caucasica.

3. Local Karachaev breed is the most frequent goat breed raised in the Republic of Kabardino-Balkaria. Saanen goats were imported to the Republic with the aim of improvement Karachaev goat population. In this regard, we did not include SNP-genotypes for other breeds, which are presented in our earlier paper [reference #18 in the revised version], because majority of Russian goat breeds (Orenburg, Altai Mountain, etc.) are bred within their local area where they were created.

The suspect has the flock of the goats of the Karachaev breed. Besides the suspect claims that he rears the Karachaev × Tur hybrids as well. However, no official /breeding recordings were available.

The using of SNP chips is a powerful tool to detect the intraspecific hybrids because hybrid individuals would be clearly shown based on PCA and Admixture results. In addition, we tested the accuracy of hybrid recognition provided by SNPs and microsatellites and found that SNPs were more suitable for this purpose (please find the reference as T. E. Deniskova, A.V. Dotsev, V.A. Bagirov, K. et al (2017). doi: 10.15389/agrobiology.2017.2.251eng). We added this paper to the References [reference #36 in the revised version] and included relevant part in the Discussion section (P10L464-467).

4. The combination of two types of markers allowed us to determine the corpus delicti in a rather cheap and accurate manner. Thus, microsatellites are suitable for determining belonging to one individual. And microsatellites showed that all the seized samples belong to one individual. SNP markers allowed us to clearly define even the ecotype of the tur to which the studied sample belonged. And for this we had to genotype just one sample.

Reviewer 2 Report

The authors combine multiple microsatellite and genome-wide SNP genotyping analyses for a wildlife forensic study.  They identified all samples from a suspected case likely belongs to a same individual which is possibly a wild Caucasian tur (Capra caucasica).  They successfully demonstrated that this strategy and approach could be widely utilized for conservation and management of wild animals.

Comments:

1) All PCR conditions and gel pictures for Table 1 should be provided as supporting or supplementary data.   

2) Detailed results for genome-wide SNP genotyping should also be provided as supporting or supplementary data. Supposedly, not 53,347 SNPs are informative for all these samples, please clearly indicate how many percent of SNPs are informative for this analysis.

Author Response

Dear Reviewer,

thank you very much for the valuable comments!

1. We added pictures of the microsatellite profiles (screenshots from GeneMapper software) as Supplementary file (FigureS1).

The PCR conditions were detailed presented in the Materials and Methods section in the main text (P4L111-121).

2.  We added the results for genome-wide SNP in the Materials and Methods section in the main text (P5L263-268) as following: «After quality control 43726 variants passed the filters (only SNPs with known position, located on autosomes with minor allele frequency - 1% and successful genotyping in at least 90% of individuals). This set of SNPs was used for multilocus heterozygosity (MLH) analysis. For PCA, Admixture and Neighbor-Net additionally LD pruning was performed (--indep-pairwise 50 5 0.5). After LD pruning 7980 SNPs were selected».

Round 2

Reviewer 1 Report

Dear authors,

thanks for your answers and for the work done following my suggestions. However, I have some aditional comments regarding your answers.

1.- As you said, microsatellites are cheaper than SNPs arrays, but I am not sure that this reason it is enough to justify that two different molecular markers have been used. For example, microsatellites genotyping needs some facilities that not all the laboratories have and in this situation and sometimes it is not easy to find a laboratory thta do this kind of analysis while it is very easy to find a company that offers SNP genotyping services.

However if there is acessible microsatellite genotyping analysis, it is chepaer and also, in the majority of the situations, you can get the same results using SNPs and microsatellites.

Also,I think that microsatellites are better markers than SNPs to achieve if a sample has DNA from differnet indivinduals, for example, that this could be a real situtation in forensic cases.

However I agree thta from a general point of view, microsatellite genotyping is chepar than SNP array analysis, so perhaps a paragraph could be aded in the discussion section just to comennt it, justyfing that microsatellites were used first to confimr the single origin of the sample because are cheaper.

I think that the PI estimated add value to the article. It is a pity that you do not have the PI for the tur population, because it is the most important PI parameter in your analysis. In addition I am not sure if the sentence: "Statistical power of the chosen set of microsatellite loci was tested by estimation of Probability of Identity (PI) or Match Probability in random populations of goats" beacuse there are not random populations, are goat breeds that are important for the analysis because are so close to the popualtions analysed or even are the populations analysed. So the pargraph could be changed by adding the number of goat breeds analysed.

3.- I agrre with the authors that SNPs could be more powerful to detect Hybrids. However, microsatellites and SNPs can assign an anonymous sample to the wild or goat breed if the sample is not an hybryd, and sometimes the power is similar for both markers. But if the sample is an hybrid SNPs are more powerful. Also, I am not sure if hybrids samples can change the results because if there are different hybrids in the goat population (50 wild:50 goat bred, 75:25, 90:10,...) the separation among the wild popualtions and the goats breeds will be not clear in a PCA adn as a consequence could difficult the assignment of a sample to the wild or goat breed population.

So, pehaps a paragraph could be added in the discussion section just discussing it, if the authors think that it is appropiate.

Furtheremore, I think that the remaining questions that I did have been suficciently answered by the authors.

Regards.

Author Response

Dear Reviewer,

Thank you for your consideration and for valuable comments!

  1. We agree that it might not be easy to find specialized laboratory, which performs microsatellite analysis for animal forensics purposes. Nevertheless, in Russia, SNP genotyping services based on using DNA chips for animals is only in beginning.

In our institute, we have both facilities for microsatellite analysis and SNP genotyping. And the cost for microsatellite genotyping is about 8$ US per sample and 64$ US is the price for SNP-genotyping using Illumina Goat SNP50 BeadChip per sample.

We added the paragraph in the Discussion (P9 L270-272).

Regarding the PI estimates, we agreed with the comment and changed the relevant paragraphs by adding the number of goat breeds (P4 L125-129, P5 L179-183).

  1. Thank you very much for comment! We will be more precise in the cases of hybrids in our future works. We rephrased the paragraph (P10L315-318).
